# Host Restriction Factors Modulating HIV Latency and Replication in Macrophages

**DOI:** 10.3390/ijms23063021

**Published:** 2022-03-11

**Authors:** Isabel Pagani, Pietro Demela, Silvia Ghezzi, Elisa Vicenzi, Massimo Pizzato, Guido Poli

**Affiliations:** 1Viral Pathogenesis and Biosafety Unit, San Raffaele Scientific Institute, Via Olgettina n. 58, 20132 Milano, Italy; pagani.isabel@hsr.it (I.P.); ghezzi.silvia@hsr.it (S.G.); vicenzi.elisa@hsr.it (E.V.); 2Human Immuno-Virology Unit, San Raffaele Scientific Institute, Via Olgettina n. 58, 20132 Milano, Italy; pietro.demela@fht.org; 3Department of Cellular, Computational and Integrative Biology, University of Trento, 38123 Trento, Italy; massimo.pizzato@unitn.it; 4School of Medicine, Vita-Salute San Raffaele University, Via Olgettina n. 58, 20132 Milano, Italy

**Keywords:** HIV, macrophages, MDM, restriction factors, transcription factors, macrophage polarization

## Abstract

In addition to CD4^+^ T lymphocytes, myeloid cells and, particularly, differentiated macrophages are targets of human immunodeficiency virus type-1 (HIV-1) infection via the interaction of gp120Env with CD4 and CCR5 or CXCR4. Both T cells and macrophages support virus replication, although with substantial differences. In contrast to activated CD4^+^ T lymphocytes, HIV-1 replication in macrophages occurs in nondividing cells and it is characterized by the virtual absence of cytopathicity both in vitro and in vivo. These general features should be considered in evaluating the role of cell-associated restriction factors aiming at preventing or curtailing virus replication in macrophages and T cells, particularly in the context of designing strategies to tackle the viral reservoir in infected individuals receiving combination antiretroviral therapy. In this regard, we will here also discuss a model of reversible HIV-1 latency in primary human macrophages and the role of host factors determining the restriction or reactivation of virus replication in these cells.

## 1. Introduction on Macrophages as Targets of HIV Replication

The *Lentiviridae* genus of retroviruses is known to target mononuclear phagocytes for their replication, although they can also infect other cell types [1]. In the case of the human immunodeficiency virus (HIV), the selection of CD4 as a primary entry receptor has evolutionarily determined an expansion of its cell tropism for a major subset of T lymphocytes with “helper” function. The consequence of HIV replication in CD4^+^ T cells is their depletion in association with a profound cytopathic effect, including the formation of large syncytia in vitro. In vivo, HIV infection results in the progressive depletion of CD4^+^ T cells, leading to a state of profound immunodeficiency known as acquired immunodeficiency syndrome (AIDS) with the emergence of opportunistic infections and peculiar types of cancer resulting in the death of >95% of infected individuals if combination antiretroviral therapy (cART) is not administered; conversely, HIV infection of tissue macrophages neither causes their depletion in vitro nor in vivo, perhaps reflecting the coevolution between lentiviruses and myeloid cells [2,3]. 

HIV infection of macrophages has been largely studied in vitro upon the differentiation of human peripheral blood monocytes into differentiated cells (i.e., monocyte-derived macrophages, MDM). Although MDM have been largely interpreted as a surrogate model of the physiological differentiation pathway of myeloid cells, it has been recently demonstrated that it represents only an “emergency” pathway of cell recruitment into inflamed tissues. Indeed, most tissue-resident macrophages (TRM) derive from primordial embryonic structures and seed the peripheral tissues before the development of the vessel system, blood and bone marrow [4,5], although there are exceptions, such as gut-associated macrophages of bone marrow origin [6] and, perhaps, lung interstitial macrophages [7]. TRM play a role in tissues and organs as scavenger cells removing apoptotic bodies and maintain themselves through the release of cytokines inducing their slow homeostatic turnover [4,5]. However, in the case of infection or inflammation, circulating monocytes are recruited in the damaged tissue in response to chemokines and other chemotactic factors released by the site of infection and rapidly differentiate into MDM, therefore mixing with TRM in the orchestration of the local inflammatory, anti-microbial response. 

There is robust evidence that TRM are targets of HIV infection in vivo, as highlighted particularly in the central nervous system (CNS) in which microglia, upon infection in the absence of cART, drives the development of a deadly encephalitis associated with a clinical condition known as AIDS-associated dementia [8]. HIV infection of macrophages has been also shown in other tissues and organs, as reviewed in [9] and reproduced in relevant animal models such as non-human primates (NHP) experimentally infected with the simian immunodeficiency virus (SIV) [10] and immunodeficient mice reconstituted with human progenitor cells before HIV infection [11].

Thus, with the caveat that in vitro infection of MDM might not accurately reflect all features of TRM, some of the main similarities and differences between CD4^+^ T cell and MDM infection can be summarized as in Table 1.

## 2. Host Cell Restriction Factors, HIV Infection and Replication

There is abundant evidence of a complex network of restriction factors (RF) already expressed in many cell types in the absence of an infection, thereby providing a state of “intrinsic immunity” [21]. A general feature is their upregulation by interferons (IFNs) that are generated in response to viral infection or vaccination. In the case of HIV, the first clear-cut evidence of “intrinsic immunity” came by studying cell lines with either “permissive” or “nonpermissive” phenotypes in terms of virus infection and replication. Permissive cell lines lacked expression of an intracellular gatekeeper belonging to the apolipoprotein B mRNA-editing catalytic polypeptide (APOBEC) family [22], namely, APOBEC3G (A3G), a cytidine deaminase targeted to proteasomal degradation by the virion-associated accessory protein Vif. In the case of viruses lacking Vif expression, A3G expressed by host cells interferes with the process of reverse transcription of the genomic viral RNA into DNA by converting the cytosines present in the minus strand into uracils, thereby resulting in the accumulation of G-to-A mutations in the plus strand. This seminal discovery inspired the paradigm that restriction factors are usually counteracted by specific viral genes as a result of a long coevolution process [23]. 

In the case of HIV, and of retroviruses in general, the goal of restriction factors is best achieved when acting before the integration of proviral DNA into host cell chromosomes in order to curtail the number of persistently infected cells unaffected by cART that contribute to the establishment of the so-called “viral reservoir”, nowadays a major obstacle in HIV eradication [2,24]. Once proviral integration has occurred, however, certain transcription factors could be also considered operationally as “restriction factors” by favoring latent vs. productive infection, as later discussed. Finally, additional restriction factors have been shown to target late steps in the viral life cycle, therefore affecting viral spreading to target cells, as in the case of BST-2/Tetherin, as later discussed.

An overall list of the main restriction factors relevant to HIV infection of macrophages and T cells and their main mechanism of action (when identified) is visualized in Figure 1 and summarized in Table 2.

## 3. Viral Proteins Counteracting Restriction Factors

Among other accessory genes encoded by HIV, Vpr, a virion-associated protein, was characterized as a relevant factor to allow for efficient virus replication in macrophages [67,68]. Its mechanism of action has been debated for several years and likely does not rely on a single modality. Experimental evidence supports an active role of Vpr as a transcriptional booster of provirus expression potentially involving its interaction with the intracellular glucocorticoid receptor [69] followed by its translocation from cytoplasm to the cell nucleus and leading to cell cycle arrest in the G_2_/M phase [70,71,72]. The importance of Vpr in HIV pathogenesis is supported by in vivo studies in NHP [73] and in HIV-1+ long-term nonprogressors (LTNP) infected with Vpr-defective viruses [74] as well as by a rare case of human infection in a laboratory setting [75]. It has also been suggested that Vpr could cooperate with Vif in the interaction with A3G leading to its proteasomal degradation [76]. Another target of Vpr potentially relevant for HIV infection of macrophages is Tet2 (ten eleven translocation 2), a member of the family of DNA dioxygenase that leads to cytosine demethylation [77]. Vpr-dependent degradation of Tet2 has been associated with an increased secretion of interleukin-6 (IL-6) by HIV-infected MDM enhancing virus replication by acting in an autocrine/paracrine fashion [78]. In addition, Tet2 upregulates the expression of IFN-induced transmembrane protein 3 (IFITM3), restricting virus replication in cooperation with IFTIM2 [38]. Vpr has been shown to interfere with the transcription of type 1 IFNs [79,80] as well as with the recently described RNA-associated early-stage antiviral factor (REAF) acting on early steps of the virus life cycle both in cell lines [81] and in primary human MDM [33]. Finally, Vpr was shown to interfere with karyopherin-mediated import of NF-kB and interferon regulatory factor 3 (IRF3) and inhibited cyclic guanosine monophosphate–adenosine monophosphate (cGAMP)-dependent expression of cyclic GMP-AMP synthase (cGAS), thereby impairing the innate immune response to the incoming infection [82].

In addition to A3G, other members of the APOBEC family have shown similar anti-HIV activity, with particular regard to A3C, A3D, A3F and A3H, as reviewed in [83]. Like A3G, A3F and A3H (but not other members) are also inhibited by Vif [84]. However, A3G has been reported to restrict HIV-1 to a greater extent than A3F and A3DE in both human primary CD4^+^ T Cells and MDM [84]. Although A3A was not initially included among those, we [85] and others [86,87] have reported its potential role as a restriction factor for HIV-1 infection in monocytes and macrophages, as later discussed. Of interest, the antibody (Ab)-mediated downregulation of endogenously released chemokine CCL2/Monocyte Chemotactic Protein-1 (MCP-1) leads to an IFN-independent upregulation of A3A expression [88]. In the same study, CCL2 downregulation led to an NF-KB mediated upregulation of the microRNA miR-155 that modulated the expression of several genes, including chemokines and their receptors [88].

A peculiar property of macrophages is their long survival in culture in a nonproliferating state (whereas, in vivo, tissue-resident macrophages, TRM, undergo a slow homeostatic proliferation driven by cytokines secreted in an autocrine fashion in order to persist indefinitely in the host [4,5]) caused by a physiological arrest of their cell cycle. This aspect of their biology has been recently revisited for its relevance to HIV infection and the role of restriction factors, with particular regard to SAMHD1 (SAM domain and HD domain-containing protein 1), a hydrolase processing deoxynucleotides triphosphates (dNTPs) physiologically involved in DNA repair mechanisms [13,89]. SAMHD1 acts by depleting the pool of dNTPs, the “building blocks” necessary for the reverse transcription process to synthesize viral DNA before its integration into the host cell genome. Vpx is an accessory protein of HIV-2 (that is not, however, present in the HIV-1 genome) that targets SAMHD1 for proteasomal degradation, thus allowing for efficient virus replication [39,40]. As HIV-1 is devoid of Vpx, it has been highly debated whether a similar mechanism would be carried out by other accessory viral proteins. In this regard, Ferreira and colleagues have reported that, although in the absence of cell division [13], macrophages in the G_0_ phase express p21/Waf1 (previously shown to represent a negative regulator of virus replication in macrophages [51]) together with high levels of SAMHD1, thereby resulting in a highly restricted state for virus replication. A switch to G_1_ has been associated with the downregulation of p21/Waf1, increased expression of cyclin-dependent kinase 1 (CDK1) and inactivation of SAMHD1 by phosphorylation, leading to increased dNTP levels and the unleashing of virus replication. This pathway was reverted by genotoxic stress and response to “danger signals” that induced a back-transition from G_1_ to G_0_, as reviewed in [13]. The inhibitory activity of SAMHD1 on HIV-1 replication has been also reported in the case of the infection of resting CD4^+^ T lymphocytes [90]. 

A well-described feature of HIV-1 replication in macrophages is the lack of induction of a robust type 1 IFN response upon infection, likely explained by the activity of the TREX1 exonuclease [42,91]. TREX1 downregulates the synthesis of IFN triggered by HIV-1 infection in both CD4^+^ T cells and macrophages; HIV-1 DNA accumulates in the cytoplasm in cells in which *TREX1* is inhibited by RNA-mediated interference, although a modest induction of IFN-induced genes (ISG) can be observed upon infection [92,93]. In the absence of a specific cytosolic sensor, it has been hypothesized that the interaction with HIV-1 virions with the plasma membrane would be sufficient to trigger a canonical activation of a stimulator of interferon genes (STING)-dependent pathway involving IRF-3 [94,95]. This pathway is activated by the transfer of cGAMP from HIV Env-expressing cells to macrophages, thereby inducing a STING-dependent IFN response protective against infection [96]. In addition, a second wave of ISG expression has been reported to occur a few days after infection following proviral integration and synthesis of the viral regulatory protein Tat, as reviewed in [97]. 

The IFN-inducible human myxovirus resistance protein B or 2 (MxB/Mx2), related but distinct from MxA, has been recently described as a restriction factor for HIV-1 infection by acting after the completion of the reverse transcription process but before proviral integration, likely through interaction with the capsid-cyclophilin A complex [44,45,98,99]. Recently, Mx2/MxB expression was required for the restrictive activity of SAMHD1, again HIV, but not SIV, in the monocytic cell line THP-1 and in primary human MDM [100]. The antiviral activity of MxB/Mx2 has been also reported in the SupT1 CD4^+^ T cell line, as reviewed [101].

Once integrated as proviral DNA, the expression of the HIV genome falls under the control of both viral and host factors influencing its transcription, RNA splicing, the export of viral mRNAs from the nucleus to the cytoplasm, translation into viral proteins and their assembly with full-length viral RNA at the plasma membrane to generate new progeny virions. The role of negative regulators of proviral transcription (Table 2) will be discussed later.

The last steps of the viral life cycle, namely, the budding and release of new virions, are targets of modulation by several RF, as earlier described in the case of Tetherin/BST-2, an IFN-inducible tetraspanin that keeps the virions stuck at the cell surface without being released; its action is counteracted by the viral accessory protein Vpu that promotes its degradation [59,102]. In macrophages, Tetherin inhibits HIV release by retaining nascent particles in assembly compartments, named “virus containing compartments (VCC)” and can also restrict the transmission of HIV by intercellular contacts between macrophages and T cells [103]. In addition to IFN, the viral accessory protein Nef has been reported to increase the levels of Tetherin expression in macrophages [57]. Conversely, the role of Thetherin in T cell infection is more controversial, as discussed [104].

Proteins of the IFITM family (in particular, IFITM1, 2 and 3) interfere with the release of new virions by insertion into the Env of nascent virions, thereby impairing cell fusion in models of cell-to-cell viral spreading [105,106]. Of interest, their restriction of virus replication has been reported to be particularly effective in macrophages in comparison to T cells [107].

Two recently identified restriction factors, membrane-associated RING-CH 8 (MARCH8) and guanylate binding protein-5 (GBP-5), expressed in macrophages, target HIV-1 gp120 Env, therefore playing a role in the final phases of the budding and release of new progeny virions [61,63]. Members of the MARCH family of E3-ubiquitin ligases are under the control of IFN and were previously shown to downregulate several host cell proteins from the plasma membrane, thereby limiting the levels of HIV-1 Env incorporation into budding virions; therefore, this results in decreased infectivity. In particular, MARCH8 is endogenously expressed by MDM and dendritic cells, whereas its KO results in a significant boost of virion infectivity [63]. The CRISPR-mediated depletion of MARCH8 in MDM, the cells of which exhibit high endogenous expression of this E3 ligase, markedly enhanced the infectivity of the viruses produced from these cells [62]. Similarly, MARCH1 and MARCH2 likely interfere with Env incorporation into virions [64], as reviewed in [97]. It is currently unknown whether similar activities are exerted in CD4^+^ T cells.

Serinc3 and Serinc5 are host molecules that decrease virion infectivity by interfering with the fusogenic properties of HIV-1 Env glycoproteins, whereas the accessory viral protein Nef prevents their incorporation into nascent virions by acting on the plasma membrane [25,26]. Their effect has been mostly studied in cell lines; however, although they seem to be dispensable for virus replication in activated CD4^+^ T cells, they are particularly effective in primary MDM, although with significant inter-donor variability [108]. Serinc5 expression is upregulated during the differentiation of monocytes into MDM [109] and its incorporation into virions has been linked to the upregulation of pro-inflammatory cytokines, including tumor necrosis factor-α (TNF-α), interlukin-6 (IL-6) and others, an effect that was also prevented by Nef [108]. Of interest, the release of pro-inflammatory cytokines by MDM was stimulated by Serinc5 in synergy with the CCR5-antagonist Maraviroc (that prevents infection by blocking virions bound to CD4 on the plasma membrane) [110]. Indeed, Serinc5-expressing virions showed a greater susceptibility to inhibition by either Maraviroc or anti-HIV neutralizing Abs [111]. Thus, in addition to its direct antiviral effect, Serinc5 incorporation into virions may serve as “danger signal” expressed by infected cells finalized to counteract the infection. In contrast, virion incorporation of Serinc5 only moderately impaired virion infectivity in activated CD4^+^ T lymphocytes and it was not associated with altered innate immune recognition [108].

A restriction factor that lately emerged as being capable of influencing the infectious capacity of HIV is the IFN-inducible cholesterol 25-hydroxylase (CH25H), an enzyme that catalyzes the synthesis of oxysterols, mediators of several processes, including inflammation and immune activation [28,29]. In addition to HIV, this ER-associated enzyme can limit the infectivity of several enveloped viruses, including Ebola, Influenza A viruses and poliovirus, an RNA virus lacking the envelope as reviewed in [112]. Overall, oxysterols are believed to restrict viral infections at the levels of cell entry by interfering with the fusion of the virion-target cell membranes, as discussed in [112]. The negative effect of CH25H on HIV-1 replication was observed in both MDM and IL-2 stimulated PBMC infected in vitro [113], a condition in which both T cells and differentiating monocytes support HIV-1 replication [114]. No information on the potential effect of CH25H on HIV-1 infection of human CD4^+^ T cells is currently available, although the original description of its antiviral activity was reported against mitogen-stimulated primary lymphocytes of monkeys infected with SIV [115]. 

The mannose receptor (MR) expressed by macrophages plays a fundamental role as an extracellular sensor of bacterial and fungal invasion by promoting their phagocytosis. *Mycobacterium tuberculosis* countermeasures include the synthesis of lipoarabinomannan that binds to and downregulates the MR leading to the inhibition of a protective inflammatory response, as reviewed in [116]. In addition, the MR can serve as an entry receptor for Dengue virus and can also interact with HIV-1 virions that may facilitate trans-infection of CD4^+^ T lymphocytes [117] like other cell surface molecules, including dendritic cell-specific ICAM-grabbing non-integrin (DC-SIGN) [118] and the integrin α4β7 [119]. Interestingly, MR is downregulated by the HIV-1 accessory proteins Vpr [65] and Nef [66]. As the MR destabilizes gp120 Env expression on the MDM cell surface, the Vpr dual targeting of MR and IFITM3 may ultimately favor the capacity of infected MDM to spread the infection, as discussed in [65].

A synthetic view of main restriction factors affecting virus replication in macrophages is shown in Figure 1.

## 4. Pharmacological Targeting of HIV Proteins

Given their important pathogenetic role by counteracting host cell RF, HIV accessory proteins have become natural targets for the development of pharmacological agents that could synergize with cART or become relevant for “HIV Cure”-related research [120]. Although an in-depth discussion of this topic is beyond the scope of the present articles, some key information is summarized in Table 3.

Several inhibitors are being studied in the perspective of potentiating antigen presentation to CD8^+^ cytotoxic T lymphocytes by reverting Nef-induced downregulation of MHC class I, as discussed [123]. Vif inhibition of the cytidine deaminase activity of APOBEC family members has been selected as a target of pharmacological screening by mutagenizing these restriction factors [125]. In consideration of the multiple levels of Vpr interference with cell function, including metabolism and cell cycle regulation, there has been an active search of inhibitors without thus far succeeding in the identification of specific candidates that could be tested in a clinical setting, as discussed [126,127]. Concerning BST-2/Tetherin, a high-throughput screening has identified candidate inhibitors capable of interfering with Vpu-mediated inhibition [124]. In the case of HIV-2-restricted Vpx, midostaurin was identified as possessing both pro- and anti-viral effects in macrophages by, on the one hand, blocking the cell cycle in the G_2_/M phase, thereby preventing SAMHD1 phosphorylation and activation; on the other hand, in the absence of SAMHD1, midostaurin showed enhancing effects on proviral transcription and viral replication [128].

In addition to accessory genes, the regulatory proteins Tat and Rev are also being targeted for drug development. Didehydro-Cortistatin A (dCA) is a prototypic agent used in “HIV Cure” strategies of “block and lock” by interfering with Tat-dependent proviral transcription as opposed to the most exployted “shock/kick and kill” to tackle the viral reservoir resistant to cART [24,121]. In the case of Rev, an extensive high throughput screening has identified different types of inhibitors, including agents interfering with the protein itself, or with its interaction with Rev-regulatory regions (RRE), or with Rev interaction with the nuclear Exportin 1, as reviewed in [122]. 

## 5. Macrophage Polarization to a Pro-Inflammatory Mode: A Model of HIV-1 Restriction

In addition to a profound revisitation of macrophage ontogenesis, their biology has been reinterpreted beyond the classical view of a cell responding to pro-inflammatory signals according to a simple “on-off” modality. An “M1/M2” paradigm of functional polarization was generated in analogy to the well-defined Th1/Th2 established modality of diversification of CD4^+^ T helper cells. Classically activated M1 macrophages are induced by pro-inflammatory signals, including bacterial endotoxin and cytokines such as IFN-γ and TNF-α, whereas M2 macrophages are “alternatively activated” by anti-inflammatory and immunoregulatory signals such as IL-4, IL-10 and others [129]. M1 macrophages contribute to the inflammatory process and, in general, exert anti-microbial and anti-cancer effects, whereas M2 macrophage polarization displays anti-inflammatory effects together with tissue-regenerating activities, including neoangiogenesis, for which they are usually considered a functional condition favoring cancer growth while exerting variable effects in terms of antimicrobial activities, as reviewed in [129]. Of note is the fact that, unlike Th1/Th2 lymphocytes, macrophage polarization is a transient condition with cells returning to their basal state a few days after the polarizing signals have been withdrawn [130]. M1 polarization has been reported to be associated with the modulation of drug efflux transporters [131] that might bear consequences in terms of susceptibility to anti-HIV agents. This dichotomous view has been tempered by considering “M1” and “M2” as the two extremes of a spectrum of functional profiles [132].

We have originally investigated the implications of M1/M2 polarization for HIV-1 infection of primary human MDM by preincubating them with IFN-γ and TNF-α or with IL-4, in order to induce M1 or M2 polarization, respectively; the cytokines were then removed before infection [130]. Quite unexpectedly, both M1 and M2 polarization led to reduced levels of virus replication, although with different profiles. M1-MDM showed a more robust inhibition of virus replication (ca. 90% vs. control, unpolarized cells), whereas M2-MDM decreased HIV-1 production by ca. 50% although M2-induced inhibition of virus production lasted longer than that caused by M1 signals that vanished ca. 3 days after the removal of the cytokines [130,133]. M1 polarization was associated with a profound downregulation of CD4 from the plasma membrane together with an upregulated secretion of some CCR5-binding chemokines. Therefore, we initially proposed that M1 polarization induced a potent, yet partial restriction of viral entry, as supported by the quantitative analysis of cell-associated HIV DNA [130]. Concerning M2-MDM, we further demonstrated a significant role of DC-SIGN in the restricted patterns observed [118]. These general profiles of HIV replication containment have been independently confirmed by other investigators [134].

The M1 restriction was also observed bypassing the cell entry step using a VSV-G pseudotyped virus [85], suggesting a model whereby M1 polarization imposes different “hurdles” to virus replication both at the levels of viral entry and at one or more post-entry steps [133]. A clear-cut upregulation of APOBEC-3A (A3A) in M1, but not in M2, MDM that returned to the levels of monocytes was observed [85]. This observation, together with independent reports [86,87], suggests that A3A could be involved in the overall restricted profile of HIV-1 replication typical of M1-MDM (Figure 2A).

## 6. Exploiting M1 Polarization to Generate a Model of Reversible HIV-1 Latency in Primary MDM

We next observed a very reproducible pattern whereby repolarization with M1-cytokines of HIV-1 infected M1-MDM (an experimental condition that we defined as “M1^2^-MDM”) drove virus replication to nearly undetectable levels (in the absence of cytopathic effects) (Figure 2B). This quasi-silent pattern of HIV expression was not correlated with the lack of activation of STAT1 and NF-kB in these cells, whereas other factors known to act as repressors of proviral transcription, namely, TRIM22 and CIITA, were also upregulated [53].

TRIM22, also known as Staf50, is an ISG whose expression is profoundly upregulated by IFN stimulation of different cell types [47]. Our group described it as the key factor differentiating U937 cell clones with a restrictive phenotype in terms of supporting HIV-1 replication (“Minus clones”) in comparison to those fully permissive clones (“Plus clones”) [48]. TRIM22 does not possess a DNA-binding domain and acts indirectly by preventing the binding of Sp1, a positive transcription factor constitutively expressed by many cell types, to the core promoter of the HIV-1 provirus [135].

Class II transactivator (CIITA) was originally described to be the key transcription factor for MHC Class II antigen expression under the control of IFN-γ, as reviewed [56]. It was then demonstrated to play a significant role in the inhibition of both HTLV-1/HTLV-2 transcription, whereas it competed with the HIV regulatory protein Tat for binding to P-Tefb (a complex formed by Cyclin t1 and CDK9), resulting in the downregulation of proviral transcription and expression also in monocytic cells [136]. 

Independently of our studies, other investigators have exploited the hypothesis that the functional polarization of macrophages into M1 or M2 cells might bear implications for their susceptibility to HIV infection and/or restriction of viral replication. Schlaepfer and colleagues reported that M1-MDM show an increased expression of several toll-like receptors (TLR), whereas their stimulation by TLR ligands, particularly in the case of TLR2, 3, 4 and 5, leads to a further restriction of HIV replication occurring mostly before proviral integration and independently of IFN-α expression [134]. In the same study, TLR stimulation lead to the upregulation of several RF in M1-MDM [134]. Furthermore, a restrictive phenotype analogous to the M1 condition was described to occur upon exposure of MDM to commensal bacteria that lead to a latent infection driven by a type I IFN response [137].

Cocultivation of M1^2^-MDM with allogeneic PBMC stimulated by the mitogen phytohemagglutinin (PHA), or incubation with their culture supernatant, potently reverted virus infection into a productive one [53]. Rescue of HIV replication in M1 and M2 MDM has been independently reported upon cell-to-cell contact with CD4^+^ T regulatory cells [138].

Wong and colleagues [139] developed an M1/M2-MDM infection model based on the use of an infectious virus expressing the enhanced green fluorescent protein (eGFP) observing, on the one hand, that infection leads to a spontaneous state of latency in some cells and, on the other hand, confirming the overall partial restriction of virus replication that we had earlier reported [130]. Of interest, they observed that bryostatin, a pharmacologic agent activating protein kinase C, and vorinostat, an inhibitor of histone deacetylases (HDAC), but not panobinostat, another pan-HDAC inhibitor [140], were able to reverse the restriction in latently infected MDM. In addition, they observed that M2 polarization had a divergent effect on HIV-1 infection of MDM in that it caused inhibition of the initial virus replication, but it enhanced effect viral reactivation in latently infected cells [139].

Thus, restimulation of M1-MDM with different signals represents a robust model to investigate which factors can affect, either positively or negatively, HIV-1 latency and expression in myeloid cells (Figure 2B).

TRM displaying a mix M1/M2 phenotype, and not CD4^+^ T cells, have been shown to represent the viral reservoir in urethral tissue of HIV-infected men receiving cART and undergoing gender reassortment [17], whereas the existence of M1-polarized macrophages exhibiting a restricted profile compatible with our in vitro observations was demonstrated in vivo in the case of decidual macrophages [141].

## 7. Conclusions

HIV-1 infection of CD4^+^ cells primarily involves a prominent subset of T lymphocytes and myeloid cells, the latter of which encompass very different cell types, such as myeloid dendritic cells, circulating monocytes, eventually extravasating to become MDM in inflammatory conditions, and TRM that acquire very distinctive features according to the anatomical site (from Kupffer cells in the liver to the microglia in the CNS) [4,5]. Although the prominent role of latently infected CD4^+^ T cells in establishing and diversifying the HIV reservoir of cells carrying replication-competent proviruses has been well characterized, growing evidence indicates that myeloid cells, and TRM in particular, could also contribute significantly to this unsolved issue preventing the eradication of HIV-1 infection from cART-treated individuals. The identification of specific extracellular and intracellular factors influencing the susceptibility of TRM to becoming targets of either latent or productive infection could be a crucial goal in order to define effective strategies aiming at the curtailment of the HIV-1 reservoir or at its definitive silencing.

## Figures and Tables

**Figure 1 ijms-23-03021-f001:**
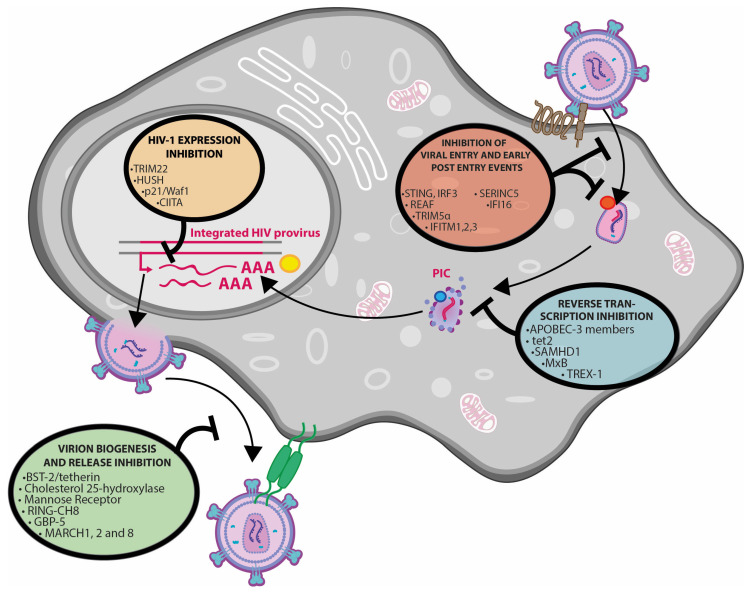
Restriction factors interfering with HIV-1 infection of macrophages. PIC: Pre-Integration Complex. See Table 2 for details.

**Figure 2 ijms-23-03021-f002:**
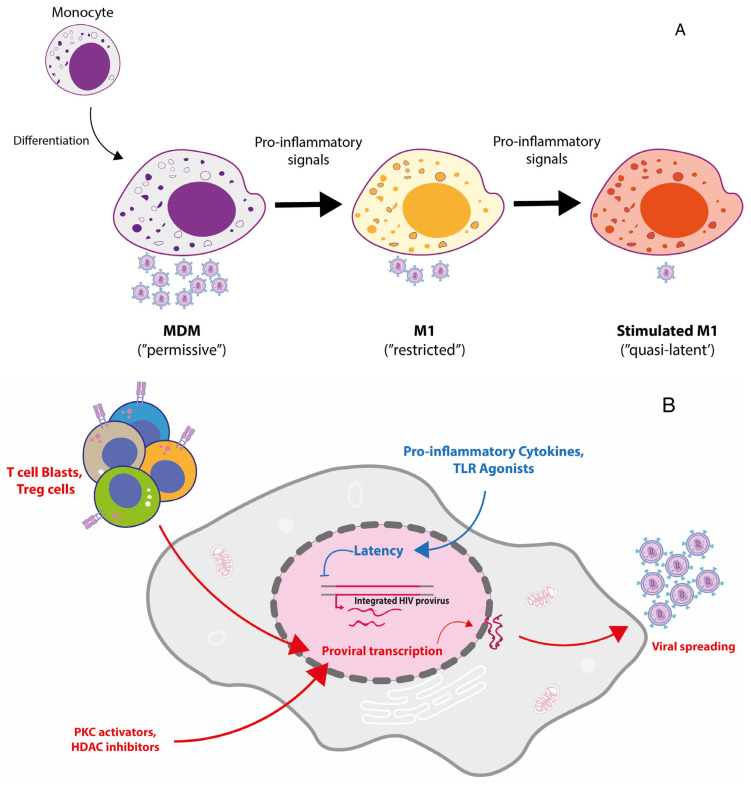
M1-MDM: a model of HIV-1 restriction suitable to be exploited as model of reversible HIV-1 latency. (**A**) Differentiation of circulating monocytes into MDM leads to their productive infection by HIV-1 (“permissive phenotype”), whereas their polarization into M1 cells induces a significant reduction in virus replication (“restricted phenotype”). Further stimulation of M1-MDM by different proinflammatory signals leads to an even greater reduction in their capacity to support virus replication approaching a state of proviral latency. (**B**) Incubation of restimulated M1-MDM with either T cell blasts, T regulatory cells or with selected pharmacological agents leads to reversal of proviral latency in MDM and viral spreading.

**Table 1 ijms-23-03021-t001:** Similarities and differences between CD4^+^ T cell and MDM infection in vitro and in vivo.

	CD4^+^ T Cells	Macrophages	Refs.	Notes
Entry receptors	CD4, CCR5, CXCR4	CD4, CCR5, CXCR4	[2]	Although macrophages express CXCR4 productive infection is usually associated with HIV CCR5 use
Cell proliferation	Yes	No	[12,13]	
Cytopathic effect, cell depletion in vitro	Yes	No	[9]	
Cytopathic effect, cell depletion in vivo	Yes	No	[2,9]	CD4 T cell depletion in vivo is likely the result of different processes in addition to virus-induced cytopathicity
Main pathogenetic consequence	Profound immunodeficiency, opportunistic infections, cancer	Tissue pathology, brain infection (encephalitis)	[2,14]	
Virus budding and release	Plasma membrane only	Plasma membrane and VCC	[15,16]	VCC are defined as invaginations of the plasma membrane connected or not to the cell surface and the extracellular environment
Role as viral reservoirs in cART-treated individuals	Well-demonstrated in the case of latently infected “resting memory” cells	Strong evidence in support of TRM	[17,18,19,20]	TRM are credited with a longer ½ life than MDM

**Table 2 ijms-23-03021-t002:** Main restriction factors and other inhibitory molecules curtailing HIV infection or replication in human macrophages and T cells.

Restriction Factor	HIV Life Cycle Step Affected	Mechanism of Action	Counteracting Viral Protein	Key Refs	Notes
SERINC3/5	Viral entry	Prevention of virion-cell fusion	Nef	[25,26]	
IFITM1, 2, 3	Viral entry	Incorporation into nascent HIV-1 virions and prevention of cell fusion	Vpr	[27]	
CH25H	Viral entry	Prevention of virion-cell fusion		[28,29]	
STING	Post-entry events	Induction of the IFN response	Vpr, Vpx	[30,31,32]	HIV-2 only
REAF	Early post-entry events	Unclear/unknown	Vpr	[33]	
TRIM5α	Early post-entry events	Degradation of the incoming viral capsid		[23,34,35,36,37]	Human TRIM5α prevents animal lentivirus infection, whereas cyclophilin A prevents its binding to HIV in human cells
APOBEC3 members	Reverse transcription	C to A hypermutation	Vif	[23]	
Tet2	Reverse transcription	Cytosine demethylation	Vpr	[38]	
SAMHD1	Reverse transcription	Depletion of dNTP pool	Vpx	[39,40]	SAMHD1 gene is involved in the Aicardi Goutières Syndrome
TREX-1	Reverse transcription	prevention of IFN induction		[41,42]	TREX1 gene is involved in the Aicardi Goutières Syndrome
IFI16*	Reverse transcription	Induction of IFN response		[43]	IFN-inducible protein 16 interacts with single stranded HIV DNA
Mx2/MxB	Post-reverse transcription	Interaction with PIC		[44,45,46]	PIC: Pre-Integration Complex
TRIM22	Integrated provirus	Transcriptional repression		[47,48]	Inhibition mediated by interference with Sp1
NF-kB1 (p50) homodimers	Integrated provirus	Transcriptional repression		[49,50]	NF-kB1 can form heterodimers with C-terminally truncated STAT5 to repress proviral transcription
p21/Waf1	Integrated provirus	Transcriptional repression		[51,52]	
CIITA	Integrated provirus	Transcriptional repression		[53,54]	Class II transactivator, also repressed HTLV-1/2 Tax transcriptional activity
HUSH Complex	Integrated provirus	Transcriptional repression	Vpx, Vpr	[55,56]	HUman Silencing Hub
BST-2/Tetherin	Budding and virion release	Prevention of virion release from plasma membrane	Vpu (Nef)	[57,58,59,60]	IFN-α stimulation upregulates BST-2/tetherin expression. In addition, Tetherin can trigger NF-kB activation after binding of Vpu-defective HIV
GBP-5	Budding and virion release	Prevention of envelope incorporation into virions	Vpu	[61]	
MARCH1, 2 and 8	Budding and virion release	Prevention of envelope incorporation into virions		[62,63,64]	
Mannose Receptor	Budding and virion release	Prevention of envelope incorporation into virions	VpR, Nef	[65,66]	

**Table 3 ijms-23-03021-t003:** Pharmacological targeting of HIV accessory proteins.

HIV Protein	Correlated RF	Pharmacologic Inhibitor	Key References
Tat	n.a.	Didehydro-Cortistatin A	[121]
Rev	n.a.	Several molecules	[122]
Nef	SerinC3/C5	Several molecules	[123]
Vpu	BST-2/Tetherin	Several molecules	[124]
Vif	APOBEC family members		[125]
Vpr	Several (see Table 1)		[126,127]
Vpx	SAMHD1		[128]

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
