# Peer review of "Host Restriction Factors Modulating HIV Latency and Replication in Macrophages"

_ijms, 2022, doi:10.3390/ijms23063021_

Round 1

Reviewer 1 Report

The manuscript entitled "Host Restriction Factors Modulating HIV Latency and Replication in Myeloid Cells" smoothly describes the main restriction factors involved in HIV infection, specifically in myeloid cells. This review is well written and include an extended information of the modulation of HIV latency, either by host restriction factors and viral proteins. This data could provide new insights in the fight against HIV. However, there are some minor points which can be improved for publishing:

There are some several long sentences that could be improved if separated.

Line 246 lacks a reference.

Line 322, the sentence needs to be rewritten

Update the references, 60% are older than 5 years.

Author Response

The manuscript entitled "Host Restriction Factors Modulating HIV Latency and Replication in Myeloid Cells" smoothly describes the main restriction factors involved in HIV infection, specifically in myeloid cells. This review is well written and include an extended information of the modulation of HIV latency, either by host restriction factors and viral proteins. This data could provide new insights in the fight against HIV. However, there are some minor points which can be improved for publishing:

  1. There are some several long sentences that could be improved if separated. R: OK, text rewritten.
  2. Line 246 lacks a reference. R: OK, references inserted
  3. Line 322, the sentence needs to be rewritten. R: the whole text of this section has been revised according to the indications of reviewer n. 2 and essentially condensed.
  4. Update the references, 60% are older than 5 years. R: the reference section has been significantly updated; some old sections refer to seminal studies and we believe that they are justified.

Reviewer 2 Report

This review article describes HIV infection in macrophages and the role of host cell restriction factors in limiting viral replication. It includes nice detail and discussion of the role of accessory proteins in counteracting the action of these restriction factors and goes some way towards highlighting the unique features of HIV infection in macrophages. A more critical analysis of the differences in these mechanisms in macrophages as compared to other cell types, plus an extension of the observations to include potential therapeutic approaches to target HIV accessory factors would strengthen these initial sections. As it stands, section 4 of the article is out of place and not consistent in style or content with the rest of the review; this section requires substantial revision. Overall, the article is stylistically well written, but there are a number of grammatical and typographical errors that require attention.

Specific points:

  1. I enjoyed reading the review of restriction factors, with mention of data discussing their role in macrophages, but this section should be strengthened with a critical review of differences (or gaps in our understanding) between the actions of these restriction factors in macrophages as compared to other cell types eg T cells. This would help highlight the unique challenges and approaches for targeting HIV infection specifically in macrophages.
  2.  Discussion of how the information covered in this article could be used to inform clinical approaches to target HIV-infected macrophages should be included. Which accessory proteins may need to be inhibited? There is literature on eg Nef and Vpr inhibitors, including their activity in macrophages (eg https://doi.org/10.1371/journal.pone.0145573) which would be good to include. Apoptosis promoting agents eg SMAC mimetics (DOI: 10.1038/s41598-021-02146-w) have also been evaluated in macrophages and may be of interest to discuss.
  3. Line 41-43: It should be acknowledged that whilst embryonic-derived macrophages are predominant in many tissue eg the brain, MDM are typically the more prevalent type in other tissues such as the gut.
  4. Line 116-120: It is unclear if the CCL2 downregulation mentioned is in response to HIV infection? And does CCL2 downregulation increase A3A?
  5. Line 155-158: Clarify if these studies are evaluating MxB in HIV infection in macrophages or other cell types. Same for discussion of STING and IFR3 below.
  6. Line 206-208: This sentence is unclear – how does maraviroc impact SERINC5-mediated cytokine production?
  7. Line 208-209: Why are SERINC5-containing virions more sensitive to maraviroc and NAbs?
  8. Lines 250-302: This section reads like a primary research article and is a little out of place in a Review article. The impact of macrophage polarization per se on HIV infection is less relevant to this review of restrictions factors, so a very brief review of polarization, with reference to other relevant studies in macrophages (eg Schlaepfer et al J Virol 2014; Wong et al J Virol 2021), should be included, then this section should focus on the relevance of polarization to restriction factors (ie the impact on A3A, lines 270-275).
  9. Lines 276-297: The work described here is published, and thus should not be described extensively. Please reword to provide only sufficient information of the M12-MDM model to enable interpretation of the relevant findings in lines 297-314.
  10. Line 319: There has been a recent study describing a similar model which demonstrated latent HIV infection in MDM and the ability of polarization to modulate HIV reactivation (Wong et al J Virol 2021). These findings should be discussed.
  11. Lines 319-333: Please remove this detailed discussion of previous findings as they are less relevant to this review of restriction factors.
  12. Conclusions: This could be refocused to highlight what is unique about HIV infection in macrophages, how restriction factors may contribute to this, and why understanding these parameters may help inform targeting of myeloid reservoirs.
  13. The article title refers to myeloid cells, but the bulk of the discussion is focused on macrophages. Consider altering the title?
  14. Line 293: Figure 2 does not support the statements made in this sentence re viral replication.
  15. Figure 2: This could be omitted.

Minor points:

There are many grammatical errors and sentences which could benefit from being rephrased. It would be useful to carefully review the manuscript for these, but some instances are listed below:

  • Line 16: “…aiming at preventing of curtailing virus replication…” Consider rephrasing to ..aiming to prevent or …aiming to curtail.
  • Line 25: Consider replacing ‘privilege” with utilise or exploit
  • Line 48: “…in the damaged tissue through in response…” into the damaged tissue in response…
  • Line 66: Change “…intracellular factors” to “intracellular restriction factors” to be more specific.
  • Line 93: Remove the first instance of HIV

  • Table 2:
    • Correct/clarify “Low levels IFN response”
    • Trim5a – has some limited efficacy against HIV-1 – rephrase to reflect this
    • Some of the MoAs could be clarified eg instead of “Env incorporation into virions” should be “Inhibits Env incorporation..”; “prevention of cell fusion’ should be ‘prevents virion-cell membrane fusion”

  • Line 102: remove ‘early”
  • Line 103: Use the plural versions listed here
  • Line 154: Uppercase I
  • Line 196: molecules (plural)
  • Line 206: Uppercase for SERINC5
  • Line 219: as reviewed
  • Line 233: comma after ontogenesis
  • Line 251: Remove )
  • Line 246: Remove “ref”

Author Response

This review article describes HIV infection in macrophages and the role of host cell restriction factors in limiting viral replication. It includes nice detail and discussion of the role of accessory proteins in counteracting the action of these restriction factors and goes some way towards highlighting the unique features of HIV infection in macrophages.

A more critical analysis of the differences in these mechanisms in macrophages as compared to other cell types…

R: we have performed a thorough analysis of the literature in order to provide comparative information between macrophages and CD4 T cells in terms of expression and relevance of RF and related inhibitory molecules, as highlighted in several parts of the text.

…plus an extension of the observations to include potential therapeutic approaches to target HIV accessory factors would strengthen these initial sections.

R: We believe that the fascinating hypothesis that some RF could become target of therapeutic approaches would require an in-depth analysis that is beyond the scope of the present article. However, we have inserted a new Table 3 summarizing some of the most promising approaches related to this hypothesis.

As it stands, section 4 of the article is out of place and not consistent in style or content with the rest of the review; this section requires substantial revision.

R: We agree with this reviewer’s criticism and we have substantially condensed this section that has also been broadened by including a discussion of similar although independent models of HIV restriction based on functional polarization of macrophages.

Overall, the article is stylistically well written, but there are a number of grammatical and typographical errors that require attention.

R: we have reread the revised text and we hope that the errors have been amended.

Specific points:

  1. I enjoyed reading the review of restriction factors, with mention of data discussing their role in macrophages, but this section should be strengthened with a critical review of differences (or gaps in our understanding) between the actions of these restriction factors in macrophages as compared to other cell types eg T cells. This would help highlight the unique challenges and approaches for targeting HIV infection specifically in macrophages. R: as mentioned above, we have performed a thorough analysis of the literature and inserted comparative comments, when possible, in terms of relevance of individual RF and related molecules in macrophages vs. T cells
  2. Discussion of how the information covered in this article could be used to inform clinical approaches to target HIV-infected macrophages should be included. Which accessory proteins may need to be inhibited?. There is literature on eg Nef and Vpr inhibitors, including their activity in macrophages (eg https://doi.org/10.1371/journal.pone.0145573) which would be good to include. Apoptosis promoting agents eg SMAC mimetics (DOI: 10.1038/s41598-021-02146-w) have also been evaluated in macrophages and may be of interest to discuss. R: We believe that the fascinating hypothesis that some RF could become target of therapeutic approaches would require an in-depth analysis that is beyond the scope of the present article. However, we have inserted a new Table 3 summarizing some of the most promising approaches related to this hypothesis.
  3. Line 41-43: It should be acknowledged that whilst embryonic-derived macrophages are predominant in many tissue eg the brain, MDM are typically the more prevalent type in other tissues such as the gut. R: a sentence and two reference have been introduced (lines 48-49) in regard.
  4. Line 116-120: It is unclear if the CCL2 downregulation mentioned is in response to HIV infection? And does CCL2 downregulation increase A3A? R: the sentence has been rewritten to clarify these questions.
  5. Line 155-158: Clarify if these studies are evaluating MxB in HIV infection in macrophages or other cell types. Same for discussion of STING and IFR3 below. R: we have now addressed this request by better specifying the implications of these factor for HIV infection in T cells and macrophages (lines 169-170 and 178-179, respectively).
  6. Line 206-208: This sentence is unclear – how does maraviroc impact SERINC5-mediated cytokine production? R: although speculative, it is likely that both Maraviroc and Serinc5-expressing virions trigger a converging cell signaling response resulting in the modulation of cytokine production.
  7. Line 208-209: Why are SERINC5-containing virions more sensitive to maraviroc and NAbs? R: although an experimental answer is not available, these three factors (Serinc5-expressing virions, Maraviroc and NAb) converge on targeting either the primary entry receptor CD4 or the co-receptor CCR5 and their combination seems to be synergistic.
  8. Lines 250-302: This section reads like a primary research article and is a little out of place in a Review article. The impact of macrophage polarization per se on HIV infection is less relevant to this review of restrictions factors, so a very brief review of polarization, with reference to other relevant studies in macrophages (eg Schlaepfer et al J Virol 2014; Wong et al J Virol 2021), should be included, then this section should focus on the relevance of polarization to restriction factors (ie the impact on A3A, lines 270-275). Lines 276-297: The work described here is published, and thus should not be described extensively. Please reword to provide only sufficient information of the M12-MDM model to enable interpretation of the relevant findings in lines 297-314. R: We agree with this reviewer’s criticism and we have substantially condensed this section that has also been broadened by including a discussion of similar although independent models of HIV restriction based on functional polarization of macrophages.
  9. Line 319: There has been a recent study describing a similar model which demonstrated latent HIV infection in MDM and the ability of polarization to modulate HIV reactivation (Wong et al J Virol 2021). These findings should be discussed. R: we thank the reviewer to point to this interesting recent paper that we now discuss, along with others indicated by the reviewer, in lines 337-342 and 325-332, respectively.
  10. Lines 319-333: Please remove this detailed discussion of previous findings as they are less relevant to this review of restriction factors. R: done!
  11. Conclusions: This could be refocused to highlight what is unique about HIV infection in macrophages, how restriction factors may contribute to this, and why understanding these parameters may help inform targeting of myeloid reservoirs. R: OK!
  12. The article title refers to myeloid cells, but the bulk of the discussion is focused on macrophages. Consider altering the title? R: we have endorsed the reviewer’ suggestion and have modified the title accordingly.
  13. Line 293: Figure 2 does not support the statements made in this sentence re viral replication. Figure 2: This could be omitted. R: Figure 2 has been completely redrawn.

Minor points:

There are many grammatical errors and sentences which could benefit from being rephrased. It would be useful to carefully review the manuscript for these, but some instances are listed below:

  • Line 16: “…aiming at preventing of curtailing virus replication…” Consider rephrasing to ..aiming to prevent or …aiming to curtail. R: according to the Webster dictionary, to aim is a transitive verb that should be follow by “at”: transitive verb 1 to be aimed at sb[campaign, product, insult, remark]  â–¸ to be aimed at doing
  • Line 25: Consider replacing ‘privilege” with utilise or exploit R: we have replaced “privilege” with “target”
  • Line 48: “…in the damaged tissue through in response…” into the damaged tissue in response…R: correction made
  • Line 66: Change “…intracellular factors” to “intracellular restriction factors” to be more specific. R: correction made
  • Line 93: Remove the first instance of HIV R: correction made
  • Table 2:
    • Correct/clarify “Low levels IFN response” R: correction made
    • Trim5a – has some limited efficacy against HIV-1 – rephrase to reflect this. R: we have updated the information of TRIM5a by better specifying it activity against HIV in human cells
    • Some of the MoAs could be clarified eg instead of “Env incorporation into virions” should be “Inhibits Env incorporation..”; “prevention of cell fusion’ should be ‘prevents virion-cell membrane fusion” R: we have made corrections in the text
  • Line 102: remove ‘early”
  • Line 103: Use the plural versions listed here
  • Line 154: Uppercase I
  • Line 196: molecules (plural)
  • Line 206: Uppercase for SERINC5
  • Line 219: as reviewed
  • Line 233: comma after ontogenesis
  • Line 251: Remove )
  • Line 246: Remove “ref”

R: corrections made, we thank the reviewer for this extensive check!

Round 2

Reviewer 2 Report

I appreciate the authors' careful attention in responding to questions and comments on the original manuscript and thank them for modifying the work in response to enhance their review article.

There are some issues and suggestions to address prior to publication:

References appear to be out of sync throughout the manuscript such that many are in the incorrect place. Please carefully review all citations.

Thank you for including the additional information in Section 4 and table 3 as suggested (note references cited in this table are also incorrect). The table is scant on details so perhaps populate with additional information from the cited articles to make the Table more meaningful, or briefly mention selected examples in text in Section 4.  I appreciate an extensive review of this literature is beyond the scope of this article.

Many thanks for restructuring section 6. Lines 230-233 still reads like a primary article and should be rephrased. The following sentence which was removed from the original version should be reinserted at line 236 to give context to content of lines 237-245:

"This quasi-silent pattern of HIV expression was not correlated with the lack of induction of STAT1 and NF-kB in these cells whereas other factors known to act as repressors of proviral transcription, namely TRIM22 and CIITA,
were also upregulated [43]."

Thank you for citing work of Schlaepfer et al and Wong et al. Consider moving discussion of Schlaepfer et al (lines 247-251) to Section 4 as it relates to infection not latency/reactivation (suggestion only). Findings of Wong et al are slightly misrepresented - effects of LRAs were investigated on latently infected MDM, not polarized MDM. Interesting finding was differential effect of M2 polarization on latency reactivation (enhanced) vs initial infection (inhibited).

The article should be accepted for publication after addressing these small issues.

Author Response

I appreciate the authors' careful attention in responding to questions and comments on the original manuscript and thank them for modifying the work in response to enhance their review article.

There are some issues and suggestions to address prior to publication:

References appear to be out of sync throughout the manuscript such that many are in the incorrect place. Please carefully review all citations.

Reply: we thank the reviewer for indicating this issue that was due to a technical problem. All the references have been carefully checked and they are (hopefully!) correct.

Thank you for including the additional information in Section 4 and table 3 as suggested (note references cited in this table are also incorrect).

Reply: same as above.

The table 3 is scant on details so perhaps populate with additional information from the cited articles to make the Table more meaningful, or briefly mention selected examples in text in Section 4.  I appreciate an extensive review of this literature is beyond the scope of this article.

Reply: we have endorsed the reviewer’ suggestion and included a brief description of the topic (lines 191-211).

Many thanks for restructuring section 6. Lines 230-233 still reads like a primary article and should be rephrased. The following sentence which was removed from the original version should be reinserted at line 236 to give context to content of lines 237-245: "This quasi-silent pattern of HIV expression was not correlated with the lack of induction of STAT1 and NF-kB in these cells whereas other factors known to act as repressors of proviral transcription, namely TRIM22 and CIITA, were also upregulated [43]."

Reply: we have endorsed the reviewer’ suggestion and reinserted the indicated sentence after removing the lines resembling the original article.

Thank you for citing work of Schlaepfer et al and Wong et al. Consider moving discussion of Schlaepfer et al (lines 247-251) to Section 4 as it relates to infection not latency/reactivation (suggestion only).

Reply: although we have considered the Reviewer’ suggestion, we believe that the discussion of the paper by Schlaepfer et al. is more appropriate in the original section.

Findings of Wong et al are slightly misrepresented - effects of LRAs were investigated on latently infected MDM, not polarized MDM. Interesting finding was differential effect of M2 polarization on latency reactivation (enhanced) vs initial infection (inhibited).

Reply: We thank the Reviewer for pointing at our attention this interesting paper. We have corrected the sentence as indicated by the Reviewer (lines 271-272) and mentioned the pattern observed with M2 polarization of infected MDM (lines 272-273).